# Nitrate Is Nitrate: The Status Quo of Using Nitrate through Vegetable Extracts in Meat Products

**DOI:** 10.3390/foods10123019

**Published:** 2021-12-05

**Authors:** Patrícia Bernardo, Luís Patarata, Jose M. Lorenzo, Maria João Fraqueza

**Affiliations:** 1CIISA—Centro de Investigação Interdisciplinar em Sanidade Animal, Faculdade de Medicina Veterinária, Universidade de Lisboa, Avenida da Universidade Técnica, 1300-477 Lisboa, Portugal; patriciarbernardo@gmail.com; 2CECAV—Animal and Veterinary Research Center, Universidade de Trás-os-Montes e Alto Douro, 5001-801 Vila Real, Portugal; lpatarat@utad.pt; 3Centro Tecnológico de la Carne de Galicia, Adva. Galicia no. 4, Parque Tecnológico de Galicia, 32900 San Cibrao das Viñas, Spain; jmlorenzo@ceteca.net; 4Área de Tecnología de los Alimentos, Facultad de Ciencias de Ourense, Universidad de Vigo, 32004 Ourense, Spain

**Keywords:** nitrate, nitrite, nitrosamines, meat products, safety, green label

## Abstract

Nitrate and nitrites are used to give the characteristic color to cured meat products and to preserve them. According to the scientific knowledge available at the moment, these compounds are approved as food additives based on a detailed ponderation between the potential risks and benefits. The controversy over nitrites has increased with the release of an IARC Monograph suggesting an association between colorectal cancer and dietary nitrite in processed meats. The trend in “clean label” products reinforced the concern of consumers about nitrates and nitrites in meat products. This review aims to explain the role of nitrates and nitrites used in meat products. The potential chemical hazards and health risks linked to the consumption of cured meat products are described. Different strategies aiming to replace synthetic nitrate and nitrite and obtain green-label meat products are summarized, discussing their impact on various potential hazards. In the light of the present knowledge, the use or not of nitrite is highly dependent on the ponderation of two main risks—the eventual formation of nitrosamines or the eventual out-growth of severe pathogens. It is evident that synthetic nitrite and nitrate alternatives must be researched, but always considering the equilibrium that is the safety of a meat product.

## 1. Introduction

The preservation of meat with added salt and subsequent drying and smoking is a common practice worldwide. This process is deeply rooted in gastronomy and consumption habits in Mediterranean countries. The meat products preserved using these processing steps are usually referred to as cured products. In the past, the nitrate was in unpurified salt increasing its preservative action. The practice of adding nitrate and nitrite to meat products was carried out for centuries, still without scientific knowledge of its use. At the end of the 1800s, research began to elucidate the importance of saltpetre as a preservative for previously pickled meat products. During the first decades of the twentieth century, several studies were conducted in the USA and Europe, resulting in a better understanding of the importance of nitrite and nitrate in meat products. During the 1950s and 1960s, nitrite and nitrate began to be used regularly by the meat industry. Since the 1920s several countries have regulated nitrite and nitrate use, but it was mainly during the 1970s that the regulation became regular, still with considerable differences between countries. These and other details on the historical perspective of nitrite use can be consulted in reference [1].

The compounds used in the past and now are potassium or sodium salts [2]. In the present works, we will refer to them just by nitrates and nitrites. In the seventies, the use of nitrate and nitrite became controversial due to data suggesting their toxicity and consequent repercussions on human health. It was then regulated by law to prevent abusive use [3]. Experts, namely the Joint FAO/WHO Expert Committee on Food Additives (JECFA) [4], and by local food safety authorities, such as the European Union “EFSA’s expert Panel on Food Additives and Flavorings” [5] or the United States “USDA Food and Drug Administration” [6] evaluated the use of nitrate and nitrite as a food additive. According to the scientific knowledge available at the moment, these compounds are approved as food additives based on a detailed ponderation between the potential risks and benefits [7]. Additionally, food additives are continuously monitored to detect eventual situations that trigger the need to reassess their risk for human health. Despite all these preventive measures associated with the use of nitrates and nitrites in meat products, the scientific community and consumers continue to have significant concerns about it [8,9].

The controversy over nitrites has increased recently with the release of an IARC Monograph from the Working Group on the evaluation of carcinogenic risks to humans, specifically for red meat and processed meat [10]. This monograph systematizes epidemiological and toxicological studies, suggesting an association between colorectal cancer and dietary nitrite in processed meats. Nitrate and nitrite alone are considered to have no or limited carcinogenic potential [11]. However, in combination with certain amines or amides, nitrite can form *N*-nitroso compounds (NOC), many of which are carcinogenic in laboratory animals [12]. The literature presents different points of view about this issue. The main criticism on the association between colorectal cancer and processed meats relies on the potential bias introduced by the questionnaire-based association between the occurrence of colorectal cancer and processed meats consumption habits [13,14,15]. Usually, consumers can hardly refer to the amounts of a particular food they ate in the past [16], and toxicological studies used a concentration of high harmful compounds, barely possible to find within processed meats [17]. Despite the doubts in the scientific community, the precaution principle of food safety should be applied here, and strategies defined to reduce the risk for consumers [18]. Processed meats are one of the targets of these strategies [19]. Still, the potential risk of several vegetables as a source of high nitrate ingestion should also be considered, as well as drinking water in some areas of the globe [20]. The nutritional value of vegetables is unquestionable, but some, mainly green leafy ones, represent a potential risk for the endogenous formation of NOC [21,22]. However, the actual diet guidelines include the recommendation to have many vegetables in the diet and exclude meat products [23,24].

Nitric oxide, and its oxidative products, nitrite and nitrate, are vital molecules in several cellular functions and physiological systems. In clinical studies, dietary nitrate consumption has been demonstrated to have numerous health benefits, especially related to improved cardiovascular function [25]. Considering the potential health advantages, the balance between the risk and benefit of using nitrite in meat products became even more challenging to achieve.

The perception of consumers about nitrates and nitrites in meat products continues to be of great concern, particularly since the trend in “clean label” products was reinforced [26]. The consumers’ nutritional and food safety illiteracy, boosted by misinformation propagated by social media [27,28], results in an illogical behavior since they fear these compounds in meat products but not in vegetables. They do not want chemical additives but embrace other foods, such as beet juice, precisely because they contain nitrate. They do not want nitrites but accept natural products that reach nitrite amounts similar to those added into meat products [29,30].

This review aims to explain the role of nitrates and nitrites used in meat products. Potential chemical hazards and consequent health risks associated with the consumption of cured meat products are described. Different strategies to replace nitrate and synthetic nitrite are summarized, and their impact on potential hazards is discussed. Some considerations are taken about consumer perceptions of green-label meat products.

## 2. The Need for Nitrates and Nitrites in Cured Meat Products: Control of the Use

Nitrates and nitrites are inorganic salts present in non-purified crystals of common salt obtained from diverse sources. These compounds have several technological roles in meat curing processes of a sensory and safety nature. Among the functions with sensory interest, the most important is probably the formation and stabilization of cured products’ characteristic pinkish-red color. Nitrite is also involved in developing the cured flavor and preventing lipid oxidation [9,31]. The contribution of nitrite to the microbiological safety of cured meat products is eventually the most important role attributed to its use. The prevention of pathogens’ multiplication and toxinogenesis, particularly *Clostridium botulinum*, is still the more powerful argument to keep using these additives in meat [32,33,34]. 

Nitrate does not have any direct technological function. It is used as a nitrite reservoir, particularly in meat products with a long curing period, since nitrate is progressively reduced to nitrite. That reduction is usually mediated by microbial enzymes, particularly from *Staphylococcaceae* and *Micrococcaceae* elements [35]. These microorganisms are usually found in the natural fermentation microbiota in dry-cured meat products without heat treatment or starter cultures [36]. Nitrite acts directly in meat products, and its effect is achieved at the beginning of the process. It is useful in the first part of the curing process of long-cured products, and, in those with short curing periods where the use of nitrate is unwise, once there is no time for the reduction [31,37]. Nitrous acid (HNO_2_) is formed from nitrite, in an acidic environment. Nitrous acid can generate its nitrous anhydride, dinitrogen trioxide (N_2_O_3_), which has its equilibrium with nitrous oxide (NO) and nitrogen dioxide (NO_2_). Nitric oxide binds to myoglobin by Fe^2+^ bonding, originating nitrosyl myoglobin that, by denaturation of the protein part of the myoglobin by heating or drying, generates nitrosyl hemochromogen (Figure 1). This pigment is responsible for cured meat products’ expected and desired color [38]. Due to sequestrating oxygen and binding the myoglobin iron, an oxidation catalyst, nitrite also has an antioxidant role. This antioxidant role prevents the generation of off-flavors, mainly from fatty acid oxidation during meat preparation and processing, in the presence of oxygen [39].

As it is possible to see in the red box in the Figure 1, colour and nitrosamine pathways are somehow connected. In the colour formation pathway, metmyoglobin plus NO can generate nitrosilmetmyoglobin pigment. On the other hand, metmyoglobin has Fe^3+^, and its combination with NO can also fall into the nitrosamine formation pathway.

The preservative effect of nitrite is primarily associated with the inhibition of *Cl. botulinum*, one of the more severe biological hazards in meat products [40]. Nitrite and the chemical species formed from its reduction can bind iron and sulfur. These two elements are present in the so-called iron–sulfur enzyme complex of several microorganisms [41]. When the NO binds to the Fe or S of the enzyme, it loses its activity. This broad group of Fe–S enzymes includes several involved in ATP synthesis by the microorganism, namely ferredoxin in *Clostridium* spp. [9]. That inhibition of ATP synthesis inhibits the growth of the pathogen, preventing the formation of toxins—the actual hazard associated with *Cl. Botulinum* [42]. This inhibition mechanism was also observed in other pathogens, namely aerobes and facultative anaerobes [43]. The inhibitory mechanisms of nitrite may also involve peroxynitrite, a structural isomer of nitrate (OHOO^−^), highly oxidative and unstable. Due to its oxidant and nitrating capacity, it can damage several cellular functions, compromising the growth and eventually the survival of the pathogen [41].

Besides the potential association with NOC formation, which is harmful to the consumer, acute nitrite toxicity must be properly prevented [7]. Pure nitrite is not sold for the meat industry once it is used in the industry in minimal amounts. It is always used in a mixture with sodium chloride to prevent accidental poisoning resulting from excessive nitrite use. Historically, it was commercialized as “Prague salt” or “curing salt” with 6.25% of nitrite in 93.5% of salt [2]. Nowadays, it is common to have commercial preparations with around 5% of nitrite or with 5% of nitrite and 5% nitrate. For specific products, it is possible to find curing salts with slightly different quantitative compositions, usually between 0.5 and 20% of nitrite [44,45].

The use of nitrate (E251, E252) and nitrite (E249, E250) is regulated by the European Union and USA law, which dictates the maximum quantity that is allowed to be added to meat product formulas and the maximum residue levels present in meat products. This information is published in full in Regulation Nº1129/2011, of the European Union [46,47,48].

Meat processing was, and still is, associated with rurality and local markets, contributing to the sustainability of the agri-food sector. Meat products are produced in large industrial systems with technological processes quite different from the traditional ones, facing consumer demand [49]. Due to the scale of production, the large commercialization circuit and the longer shelf life, products made at a large scale use nitrate and nitrite in their formulation, while in traditional, small producers it is still common to not use any chemical additive at all [50]. The addition of additives, or not, is one practice that often separates traditional from industrial meat products [51]. This addition may result in products whose characteristics are different, including chemical hazards associated. Regarding consumers’ perception, traditional products are also associated with non-industrialized practices without additive addition; therefore, they are well accepted by the consumers [52].

## 3. The Potential Health Risks Linked to the Consumption of Nitrite-Cured Meat Products

The chemical contamination of meat can occur during production, processing, preparation, treatment, transport, storage practices, or even environmental contamination [53]. Therefore, the toxic substances present in meat can be categorized according to their different origins. The categories are geochemical pollutants from the soil, mostly anthropogenic environmental pollutants, toxic metabolites of microorganisms, endogenous animal poisons, veterinary drug residues, and toxicants borne in meat during processing and storage [54]. If we analyze the different steps of cured meat products processing, the introduction of potential chemical contaminants and the generation of others could happen during the reception of meat and ingredients already contaminated and after with the addition of particular ingredients in excess in their formulation. It should be avoided by preventive measures such as the selection and control of suppliers and weight control of the ingredients, particularly chemical additives potentially harmful for consumers’ health, as nitrite is. Other chemical compounds can be generated during fermentation, curing and smoking [55]. If caution and good practices are taken into consideration, the generation of the formation of these compounds can be prevented. However, their potential presence should always be considered [9,56]. 

During the smoking step, there is the potential formation of polycyclic aromatic hydrocarbons (PAHs), especially benzo(a)pirene, which is potentially carcinogenic [9,53]. Due to their toxicity, the content of some PAHs in foodstuffs, smoked meat products included, is regulated by Regulation Nº835/2011 of the European Union [57]. Other dangerous compounds potentially related to meat products are lipid oxidation products, protein oxidation products, heterocyclic aromatic amines (HAAs), and biogenic amines (BAs) [56,58]. Biogenic amines can be harmful directly and indirectly. Histamine, tyramine and β-phenylethylamine represent a direct potential hazard for consumers’ health because of their toxicological effects, namely their vasoactive and psychoactive proprieties. Diamines, such as putrescine and cadaverine, are not toxic per se, but their presence could be a potential indirect hazard for the consumer, not only because they can intensify the absorption of vasoactive amines but also as precursors of *N*-nitroso-compounds (NOCs) [59,60]. *N*-nitrosamides and *N*-nitrosamines are NOCs of high concern in cured meat products. These compounds are generated when nitrosating agents react with nitrosatable amines or amides. Nitrosation rate depends on several factors such as the reaction mechanism, the origin and concentration of the nitrosating agent and the nitrosatable amine or amide, and the pH of the medium, among other factors such as the presence of catalyzers or inhibitors of the reaction [53,55]. Nitrosamines can be either volatile (VNA) or non-volatile (NVNA). VNA has high carcinogenic potential, while NVNA has weak or no carcinogenic potential [61]. The most important nitrosamines in cured meat products are *N*-nitrosodimethylamine (NDMA), *N*-nitrosodiethylamine (NDEA), *N*-nitrosopiperidine (NPIP), *N*-nitrosopirrolidine (NPYR), *N*-nitrosomorpholine (NMOR), and *N*-nitrosodibutylamine (NDBA) [10,53]. Of the aforementioned, NDEA has the higher carcinogenic potential [56]. 

Since 1978, IARC has classified NDEA and NDMA as probably carcinogenic to humans (Group 2A) and NPIP, NPYR and NMOR as potentially carcinogenic to humans (Group 2B) [62]. One of the main requisites for *N*-nitrosamines formation is the presence of amines. Amines are products of amino acid decarboxylation; therefore, few compounds are expected in raw meat, but processes such as maturation/fermentation/curing may increase their formation. It is important to note that only secondary amines can form stable nitrosamines. Primary amines are immediately decomposed in alcohol and nitrogen, and tertiary amines do not react. In addition, medium pH must be sufficiently low to form nitrosyl cation (NO^+^) or metallic bonds that form NO^+^ must be present, as it is possible to see in Figure 1. Secondary amines are a subgroup of BAs that are volatile, given their structure. Dimethylamine (DMA), isopropilamine (IPA), pyrrolidine (Pyrr), piperdine (Pip), and morpholine (Mor) belong to this group [63]. Secondary amine *N*-nitrosation is one of the most studied. These compounds react with most nitrosating agents at different rates, depending on structural or physicochemical factors, namely their pKa and the steric medium near the nitrogen atom [64].

The occurrence of *N*-nitrosamines in meat products is positively related to the abundance of precursors. As found during the processing, specific conditions are needed to convert the precursor in the respective nitrosamine. Nevertheless, even if the previous requisites occur, other characteristics of the meat products can make them unsuitable for nitrosamine generation. Some cured meat products usually have low water activity and an unfavorable pH that hinders the formation of nitrosamines [65].

Apart from the processing steps of curing meat products, their culinary preparation made by consumers can trigger the generation of nitrosamines. The cooking methods, particularly those at a temperature higher than 130 °C, increase the risk of nitrosamines formation. Operations such as frying or grilling meat products may increase the probability of *N*-nitrosamine formation [66].

Several authors have published studies on the occurrence of nitrosamines in meat and meat products, with different processing conditions, from different countries (Table 1). From the analysis of these results, what stands out is the heterogeneity of nitrosamines content between different meat products and even between similar ones. That heterogeneity may cause the lack of regulation for *N*-nitrosamine content in the European Union, despite EFSA’s efforts to collect information and data about this subject [67,68]. The reference levels pointed out in the literature, based on local regulations or in the scientific limits proposed, are also very variable. As it is possible to see in Table 2, the pointed variability for the allowed limit values ranged from 2 µg/kg in Estonia to 30 µg/kg in Chile. Still, there are differences in the criteria to choose which *N*-nitrosamines should be analyzed and limited by law, indicating that there is still much work to be done on establishing standard regulations for these chemical hazards in meat products.

## 4. Strategic Use of Vegetable Extracts as a Source of Nitrates and Nitrites

Some events should be considered to understand the emergence of vegetable extracts as a source of nitrates and nitrites to be used in cured meat products. In 2003, EFSA issued a scientific opinion on the effects of nitrates and nitrites in the microbiological safety of meat products, mainly confirming their biological safety benefits, but with a warning on the potential association with nitrosamines formation [44]. To mitigate the risk, regulatory modifications were made on the additional amount of nitrate and nitrite allowed in meat products to keep the level of nitrosamines as low as possible, maintaining their microbiological safety [85]. Later, the IARC evaluated the carcinogenicity of red meat and processed meat consumption. The agency evaluated the link between red and processed meat consumption with a dozen different types of cancer. In 2015, the results of this work were issued by IARC, and processed meat was classified as carcinogenic to humans, with sufficient evidence that the consumption of processed meat increases the risk of occurrence of colorectal cancer in humans (Group 1) (IARC Working Group on the Evaluation of Carcinogenic Risks to Humans 2018). The experts’ group concluded that a daily intake of 50 g of processed meat increases the risk of colorectal cancer by 18%. Since then, the consumption of processed meat has belonged to Group 1. In the same group are tobacco and alcoholic beverages consumption, but with odds ratios dramatically lower, indicating that the increase in the probability of having cancer due to processed meats is much lower than the probability associated with the above-mentioned risk factors. 

Consumers’ reluctance for chemical additives is a latent problem in the relationship between the industry and the consumer. The press release seriously aggravated consumer perception of risk with the IARC issued in 2015, mainly due to the media treatment of the information, exploring the sensationalist discourse [86]. That situation triggered a temporary crisis in this agri-food sector due to the shortness in demand and consequent economic losses [87]. The demand for foods without synthetic additives and the controversy around their need and safety increased the need to study alternatives to synthetic nitrate and nitrite in meat products [88,89]. These alternatives must reduce or fully substitute nitrite without jeopardizing meat products’ microbiological safety or sensory characteristics, particularly desirable color and flavor [13,34].

The primary purpose is to reduce the risk of nitrosamines formation in meat products, and that has been the main reason for all studies of alternatives to nitrite since the late twentieth century [40,55]. However, it is not easy to find options that fully fulfil nitrite’s technological roles. The available knowledge regarding the safety and quality of meat products, with or without this additive, is still far beyond our needs [90,91,92]. Distinct approaches have been taken into consideration by researchers, such as the simple reduction or elimination of nitrates and nitrites [8,93], or the introduction of other technologies to achieve the desired safety and sensory results, such as high hydrostatic pressure (HHP) treatment [94], protective cultures [95], and the substitution of synthetic nitrate and nitrite for natural alternatives, which includes the use of plants [26,34,96,97,98,99]. 

The use of plants, mainly plant extracts, is probably the most studied alternative. Some plant extracts have antimicrobial and antioxidant properties. Vegetables, some more than others, are nitrate rich, thus having the potential to be used in meat products’ processing. Their combination with microbial cultures that can reduce nitrate to nitrite might answer meat processing without synthetic nitrate and nitrite addition [49,88]. Several authors have recently studied this hurdle approach [91,100].

Plants, fruits, herbs, and spices have bioactive compounds, mainly phenolic compounds with antioxidant and antimicrobial properties [101,102]. These properties of phenolic compounds from plants can be helpful to reduce the exposure of consumers to nitrosamines by two mechanisms. First, these properties can be used in meat products made without or with a reduced nitrite level to mitigate the potential problems arising from the reduced nitrite. There are several pieces of evidence that phenolic compounds can help to control the growth of pathogens in meat products through many possible mechanisms such as membrane disrupting molecules, direct pH drop, and the presence of organic acid in the plant extract [101,103], as well as preventing lipid and protein oxidation [104,105,106]. Second, phenolic compounds have nitrite-scavenging capacity. The binding of nitrite in foods can help avoid the formation of nitric oxide in the acidic pH of the stomach, indirectly avoiding the endogenous formation of nitrosamines [106].

Phenolic compounds are a very heterogeneous group of chemical compounds, considering their structure, molecular weight, and chemical, physical and biological properties [107,108]. The content of these compounds differs significantly in different plants, and factors such as organ, cultivar, and growth season may influence the composition and concentration of phenolic compounds in plant extracts, consequently influencing their potential benefits in meat processing [103].

Another conceptual strategy recently tested was the chemically modified casings or active packaging to overcome the difficulties of reducing nitrite in meat products. Alizerezalu and collaborators [109] evaluated polyamide–alginate casings with nisin, ε-polylysine nanoparticle and a mixture of plant extracts from olive leaves, green tea and stinging nettle, in a frankfurter sausage made without nitrite; they observed a reduction in the microbial growth. With different objectives, Chatkitanan and Harnkarnsujarit [110,111] studied the effect of incorporating nitrite in the packaging film used in pork. This approach was carried out in fresh meat, but it might be extrapolated for meat products, contributing to the maintenance of the characteristic color of cured meat products with low levels or without nitrite.

Although efforts are being made, it is crucial to assess if plant extracts are the alternative that meets consumers’ expectations in demanding “natural” meat products and fulfils safety and quality requirements [112].

## 5. Status Quo of Green Nitrate in Meat Products and Potential Hazards Associated

The research for natural alternatives to nitrate and nitrite has been the aim of several studies, as summarized in Table 3. Vegetable extracts must be referred to in the label, but their composition does not need to be disclosed. That is why it is so appealing to the industry to use these extracts once the name of the chemical additive is not listed [9]. Different vegetable products in meat products have been applied using different plant parts, with varied pre-treatments—powder, juice, and infusion. Their application was assayed in different proportions and in different products. Celery was the first vegetable extract to replace synthetic nitrate with plant extracts rich in that compound [113]. Several studies were conducted using vegetable extracts rich in nitrate (Table 3). The vegetable extracts can be grouped according to their nitrate level: celery, cress, lettuce, spinach, and rucola (<2500 mg/100 g); Chinese cabbage, endive, leek, and parsley (1000 to <2500 mg/100 g); turnip, savoy cabbage, and cabbage (500 to <1000 mg/100 g); carrot, cucumber, pumpkin, and broccoli (200 to <500 mg/100 g); and potato, tomato, onion, eggplant, mushroom, and asparagus (<200 mg/100 g). These concentrations are expressed as the fresh weight. When the product is dried, it is possible to obtain vegetable extracts with a much higher nitrate content [91,114,115].

There are some disadvantages associated with the use of plants as an alternative to nitrate and nitrite, such as the variability in nitrate content in plant extracts, either from different plants or the same plant, undesirable color and flavor, and the reliance on nitrate reductase activity bacteria [40]. Additionally, once it is an indirect way to add nitrite to the meat products, it is necessary to assess how it can be associated with nitrosamines’ formation. Considering the theoretical assumption that phenolic compounds present in the extract could hinder the reaction between nitric oxide and reactive amines, it is possible to infer that the probability of having these compounds is lower than with synthetic nitrite. However, there are no experimental data on that relationship, and the theoretical assumption can have a bias in extrapolating for the complex medium that is a meat product. 

The main concern in the meat products industry is the generation of *N*-nitrosamines in the product. However, as previously referred to, nitrosamines can also be generated endogenously by ingesting foods rich in nitrates and nitrites. Endogenous nitrosation may occur due to the reaction of amines with nitrous acid (HNO_2_). This reaction is supported by the aqueous and acid stomach medium, with pH values between 2.5 and 3.5. Approximately half of the nitrite in the gastric fluid is in the form of nitrous acid. Nitrous acid tends to be in equilibrium with nitrosyl cation (NO^+^) and nitrogen oxides, particularly with dinitrogen trioxide (N_2_O_3_). The latter may find equilibrium with nitrogen oxide (NO) and nitrogen dioxide (NO_2_). Thus, dinitrogen tetroxide (N_2_O_4_) formation is expected to occur and its equilibrium with NO and NO_2_. All these reactions of nitrosation and nitration can create numerous compounds that are catalyzers or inhibitors of *N*-nitrosamine formation. To summarize, in the stomach, the amine nitrosating agent is dinitrogen trioxide, resulting from the proton catalyzation of two nitrous acid (HNO_2_) molecules. As a result, the nitrosating rate is highly dependent on nitrite concentration [55].

In 2006, the IARC classified the ingestion of nitrate and nitrite in conditions that result in endogenous nitrosation as probably carcinogenic to humans (Group 2A) [116]. Nitrates in vegetables might be reduced to nitrites due to the action of buccal cavity microorganisms [13]. It is also known that the ingestion of nitrate and nitrite is mainly associated with the consumption of vegetables and meat products [20]. The European Union law shows great differences between allowed levels of nitrate in vegetables and meat products. According to Regulation N°1258/2011 of the European Union, the maximum permitted level of nitrate present in vegetables varies from 200 to 7000 mg/kg [117]. On the other hand, the maximum allowed level of residual nitrate in meat products ranges from 10 to 300 mg/kg [118]. 

Another aspect that deserves consideration, regarding the use of plants as an alternative to synthetic nitrate and nitrite, is that the Standing Committee on Plants, Animals, Food and Feed (ScoPAFF) stated that the use of plant extracts with technological properties in food processing is still considered as a use of additives [119]. Therefore, food chain operators must comply with Regulation N°1333/2008, which does not predict the use of plant extracts.

## 6. Conclusions

Nitrite and nitrate can represent a risk for consumer health. It is not an immediate risk but a result of an ensemble of factors that might favor the formation of carcinogenic nitrosamines. The contribution of nitrite to the microbiological safety of cured meat products is the most important role attributed to its use. Botulism is a very severe foodborne disease that can result in death; this occurrence can be linked to meat products. Other foodborne pathogens in meat products can also produce illnesses in the consumer, as can *Salmonella* or *Listeria monocytogenes*. In light of the present knowledge, the use or not of nitrite is highly dependent on the ponderation of these two risks—the eventual formation of nitrosamines or the eventual out-growth of severe pathogens. This review highlights the main strategies being used to face the pressure to reduce nitrites in meat products. It includes those based on reducing or eliminating nitrite and replacing synthetic nitrite with “natural” nitrite through the addition of nitrate-rich vegetable extracts. There has been research on gradual reductions in nitrite addition levels to minimize the odds of nitrosamine formation. Other research groups aimed to replace the effects of nitrite on color by optimizing the formation of natural myoglobin derivatives or using red ingredients to color the nitrite-free meat products. Replacing synthetic nitrite with vegetable nitrate must be combined with a starter with nitrate reductase activity. This last strategy is problematic because there is still the presence of nitrite. From the commercial point of view, it might be valuable if, in the label, a food additive is not presented, which might be appealing to the consumer.

On the other hand, nitrite is nitrite, and thus, this might be looked upon as a misleading strategy. Some evidence allows us to believe that vegetable nitrite might be a possible method for use in the industry, once the phenolic compounds present in the extracts are expected to prevent the formation of nitrosamines. However, that theoretical framework must be further experimentally demonstrated and confirmed. 

Besides all the technical issues specifically related to the chemistry of nitrosamines formation, we believe there is an urgent need to increase communication with the consumer. Fried bacon eaten every day at breakfast is not the same risk as eating cured ham or several slices of cured sausage in a snack. Barbecuing cured sausages over live coal is not the same as boiling them in the traditional dishes of the rich gastronomy worldwide. Consumer food literacy does not allow them to understand these differences. The misinformation undermines consumers’ confidence in this segment of foods with nutritional value, unquestionable gastronomic interest, and high economic importance.

## Figures and Tables

**Figure 1 foods-10-03019-f001:**
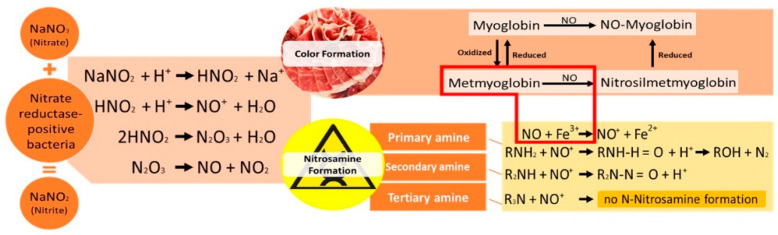
*N*-nitrosamine’s formation mechanisms.

**Table 1 foods-10-03019-t001:** *N*-nitrosamines occurrence in meat and meat products in several countries.

Product	Country	Average Concentration µg/kg									Analysis Method	Reference
		NDMA	NDEA	NDBA	NPIP	NMOR	NPYR	NMEA	NDPA	NTHZ		
Pork meat with salt	France	0.09	0.03	NA	NA	NA	NA	NA	NA	NA	GC-TEA	[69]
Pork meat	France	0.28	NA	NA	NA	NA	NA	NA	NA	NA	GLC	[70]
	Spain	1.7	ND	ND	1.0	NA	1.5	ND	ND	NA	GC-MS	[71]
Meat sausage	China	0.8	ND	0.1	ND	0.7	3.5	ND	ND	NA	GC-MS/MS	[72]
	China	2.4	ND	0.6	1.4	NA	ND	0.6	ND	NA	GC-CI-MS	[73]
	Spain *	2.2	ND	3.3	1.3	NA	ND	ND	ND	NA	GC-MS	[71]
	Spain *	2.4	ND	ND	ND	NA	1.5	ND	ND	NA	GC-MS	[71]
	Spain *	4.1	2.8	ND	1.5	NA	2.6	ND	ND	NA	GC-MS	[71]
	Spain *	3.3	2.2	1.9	1.1	NA	1.8	ND	ND	NA	GC-MS	[71]
	Spain *	4.0	1.9	ND	ND	NA	ND	ND	ND	NA	GC-MS	[71]
	Spain *	3.1	3.6	1.2	2.2	NA	1.5	ND	1.0	NA	GC-MS	[71]
	Turkey *	0.19	0.95	ND	1.05	NA	0.54	NA	0.5	NA	GCXGC-NCD	[74]
	Turkey *	0.11	0.10	0.15	0.16	NA	0.11	NA	ND	NA	GCXGC-NCD	[74]
	Turkey *	0.30	0.49	0.35	1.02	NA	0.57	NA	0.59	NA	GCXGC-NCD	[74]
	Turkey *	0.11	ND	0.19	1.49	NA	0.82	NA	0.47	NA	GCXGC-NCD	[74]
	Turkey *	ND	ND	ND	2.71	NA	1.36	NA	0.27	NA	GCXGC-NCD	[74]
	Turkey *	0.78	0.47	1.68	0.23	NA	0.18	NA	1.35	NA	GCXGC-NCD	[74]
	Italy	0,6	ND	ND	ND	ND	ND	ND	ND	NA	GC-CI/MS/MS	[75]
bacon	Germany	1.01	NA	NA	ND	NA	0.02	NA	NA	NA	GC-TEA	[76]
	Thailand	0.95	NA	NA	ND	NA	ND	NA	NA	NA	GC-TEA	[77]
	Belgium	1.6	NA	NA	0.2	NA	2.2	NA	NA	NA	HPLC-MS/MS	[61]
	Denmark	1.2	NA	NA	0.07	NA	1.4	0.5	NA	NA	HPLC-MS/MS	[61]
	China	1.4	0.4	1.7	0.4	NA	ND	0.2	0.3	NA	GC-CI-MS	[73]
Smoked bacon	France	0.25	0.97	0.17	0.11	NA	NA	NA	NA	NA	GC-TEA	[69]
Fried bacon	USA	1	0.2	0.2	ND	NA	7.1	NA	NA	1.1	GC-TEA	[78]
Ham	France	0.14	0.03	0.09	0.25	NA	0.12	NA	NA	NA	GC-TEA	[69]
	France	0.31	NA	NA	NA	NA	NA	NA	NA	NA	GLC	[70]
	Thailand	0.79	NA	NA	ND	NA	ND	NA	NA	NA	GC-TEA	[77]
	Belgium	1.5	NA	NA	0.07	NA	1.5	0.4	NA	NA	HPLC-MS/MS	[61]
	Denmark	2	NA	NA	0.04	0.08	1.2	0.2	NA	NA	HPLC-MS/MS	[61]
	Italy	0.3	ND	ND	ND	ND	ND	ND	ND	NA	GC-CI/MS/MS	[75]
	Poland	0.62	0.18	NA	NA	NA	NA	NA	NA	NA	GC-TEA	[79]
	Spain	2.6	2.4	ND	1.9	NA	3.4	ND	ND	NA	GC-MS	[71]
	China	0.6	ND	0.2	0.2	NA	ND	0.6	ND	NA	GC-CI-MS	[73]
*Bologna*	Italy	0.4	ND	ND	ND	ND	ND	ND	ND	NA	GC-CI/MS/MS	[75]
	Spain	1.5	ND	ND	0.8	NA	1.6	ND	1.2	NA	GC-MS	[71]
Black pudding	Spain	3.5	ND	3.4	2.0	NA	2.1	ND	ND	NA	GC-MS	[71]
Smoked pork brisket	France	0.53	1.6	0.45	2.9	0.25	1.2	NA	NA	NA	GC-TEA	[69]
*Prosciutto*	France	0.54	1.1	0.29	0.24	0.48	NA	NA	NA	NA	GC-TEA	[69]
	Italy	0.3	ND	ND	ND	ND	ND	ND	ND	NA	GC-CI/MS/MS	[75]
	Spain	2.0	ND	ND	1.8	NA	2.9	2.5	ND	NA	GC-MS	[71]
Other sausages	France	0.91	2.4	0.43	0.18	0.74	0.28	NA	NA	NA	GC-TEA	[69]
	Germany	0.84	NA	NA	0.03	NA	ND	NA	NA	NA	GC-TEA	[76]
	France	0.45	NA	NA	NA	NA	NA	NA	NA	NA	GLC	[70]
	Belgium	2.6	NA	NA	0.3	0.5	2.7	NA	NA	NA	HPLC-MS/MS	[61]
	Denmark	1.6	0.3	NA	0.1	NA	2.1	0.5	NA	NA	HPLC-MS/MS	[61]
	China	3.2	1.1	1.4	1.1	NA	0.7	0.4	0.7	NA	GC-CI-MS	[73]
*Salami*	France	0.45	4.6	0.56	0.17	NA	NA	NA	NA	NA	GC-TEA	[69]
	Italy	0.7	ND	ND	ND	ND	ND	ND	ND	NA	GC-CI/MS/MS	[75]
	Turkey *	ND	0.22	ND	1.44	NA	ND	NA	ND	NA	GCXGC-NCD	[74]
	Turkey *	0.3	0.28	0.10	0.73	NA	0.47	NA	0.35	NA	GCXGC-NCD	[74]
	Turkey *	0.10	0.11	ND	0.19	NA	0.14	NA	0.26	NA	GCXGC-NCD	[74]
	Turkey *	ND	ND	0.21	0.53	NA	0.37	NA	0.51	NA	GCXGC-NCD	[74]
	Turkey *	ND	0.15	0.56	0.82	NA	0.53	NA	ND	NA	GCXGC-NCD	[74]

ND: not detected; NA: not analyzed; NDMA: *N*-nitrosodimethylamine; NMEA: *N*-nitrosomethylethylamine; NDEA: *N*-nitrosodiethylamine; NPYR: *N*-nitrosopirrolidine; NDPA: *N*-nitrosodipropylamine NPIP: *N*-nitrosopiperdine: NDBA: *N*-nitrosodibutylamine; NTHZ: *N*-nitrosothiazolidine; GC-TEA: gas chromatography-thermal energy analysis; GLC: gas-liquid chromatography; GC-MS: gas chromatography-mass spectrometry; GC-MS/MS: gas chromatography-tandem mass spectrometry; GCXGC-NCD: Comprehensive Gas Chromatography with Nitrogen Chemiluminescence Detector; HPLC-MS/MS: high-pressure liquid chromatography-tandem mass spectrometry; GC-CI/MS/MS: gas chromatography-chemical ionization tandem mass spectrometry; GC-CI-MS: gas chromatography-chemical ionization-mass spectrometry. * For the same study the sampling was carried out for different producers.

**Table 2 foods-10-03019-t002:** Regulation for *N*-nitrosamine content in several countries.

Country	Maximum Permitted Level (µg/kg)	Nitrosamines	Products	References
Estonia	2	Σ NDMA and NDEA	Meat sausages submitted to heat treatment	[80]
	4		Smoked meat sausages	[80]
Russia	2	Σ NDMA and NDEA	Meat sausages	[81]
	4		Smoked meat sausages	[81]
USA	10	Total volatile nitrosamines	Cured meat product (bacon)	[82]
Canada	10	NDMA, NDEA, NDPA, NDBA, NPIP, and NMOR	Cured meat	[83]
	15	NPYR	Cured meat	[83]
Chile	30	NDMA	Cured meat product (*cecinas*)	[84]

**Table 3 foods-10-03019-t003:** Studies of natural nitrite and nitrate alternatives for meat products.

Product	Country	Alternative	Objectives	Methods	Main Conclusions	Some Disadvantages	References
Fermented dried sausage	Serbia	*Kitaibelia vitifolia* extract	Evaluate the impact of nitrite’s replacement on quality characteristics.	3 formulas: control with 27 g/kg of nitrite salt; with 30 g/kg of extract; and with 12.5 g/kg of extract. Drying process: 1 day, 22 °C, 92%RH; 1 day, 20 °C, 88%RH; 1 day, 19 °C, 86% RH; 1 day, 18 °C, 82% RH; 1 day, 15 °C, 72% RH; 21 days, 15 °C, 72% RH. Smoking process: at 3rd, 4th and 5th days, 5 h; 18 °C.	*K. vitifolia* extract revealed strong antioxidant capacity and moderate antimicrobial capacity against *E. coli*. *K. vitifolia* extract’s addition did not interfere with expected physicochemical characteristics, nor with the product’s overall acceptance. The great potential of *K. vitifolia* extracts in product’s preservation during processing and refrigerated storage	Products with addition of *K. vitifolia* extract showed lower consistency.	[96]
Fermented dried sausage	Lithuania	Lyophilized vegetable powder: celery, celery juice, parsnip, and leek	Evaluate the effect on ripening processes and final product’s properties.	5 formulas: with 3% celery powder; with 3% celery juice powder; with 3% parsnip powder; with 3% leek powder; and control without addition. Drying process: 14 days. Initial temperature 24 °C, 92% RH and gradual decrease until 15 °C, 76% RH. Smoking process: cold smoking after 4th day of ripening.	The analysis of quality parameters such as pH, a_w_, LAB, coagulase-positive *Staphylococci*, and coliform revealed that the incorporation of these vegetable powders does not have a negative effect on fermentation and ripening processes. Formulas with celery powder and celery juice powder presented relatively stable color parameters during processing.	The incorporation of these vegetable powders resulted in softer products; formulas with celery powder, celery juice powder and leek were less red.	[97]
Fermented dried sausage	Italy	Grape seed with olive pomace hydroxytyrosol and chestnut extract with olive pomace hydroxytyrosol	Evaluate the effects in physicochemical, aromatic, and sensory characteristics and microbiological safety.	3 formulas: control with 30 mg/kg of sodium nitrite; with 10 g/kg of grape seed extract; with 10 g/kg of chestnut extract. Drying process: 4 days, 28 °C, 85% RH. Ripening process: 21 days, 13 °C, 70% RH.	The replacement did not affect the overall acceptability of the products. All formulas were in agreement with European regulations for *Listeria monocytogenes*, *Salmonella*, and *Clostridium botulinum*. All formulas presented a similar aromatic profile.	Nitrite’s replacement resulted in some physical characteristics differences when compared with control. Some color characteristics of products with extracts (namely a* and b* parameters) showed significant differences when compared with control.	[98]
*Chorizo*	Spain	Natural extract of citric, acerola, rosemary, paprika, garlic, oregano, beet, lettuce, arugula, spinach, chard, celery, and watercress	Evaluate the antioxidant and antimicrobial capacity of the extracts.	7 formulas: 6 with different combinations of extracts; control without extracts. All formulas were inoculated with *C. perfringens.* Drying process: 2 days, 22 ± 1 °C, 90 ± 5 °% RH; 20 days, 14 ± 1 °C, 70 ± 5 °% RH. Storage: 125 days, packed in vacuum 5 ± 1 °C, 65 ± 5 °% RH.	Rosemary extract revealed the best antimicrobial capacity. Paprika, garlic, and oregano extracts also revealed good antimicrobial capacity. Citric extract presented high antioxidant capacity. When combined, citric extract and rosemary extract have the potential to be a good alternative to the use of synthetic additives.	Citric extract presented low antimicrobial capacity. Celery extract presented lower phenolic content and lower antioxidant capacity.	[26]
Fermented dried sausage	Italy	Grape seed with olive pomace hydroxytyrosol and chestnut extract with olive pomace hydroxytyrosol	Evaluate the effect on the prokaryotic community.	3 formulas: control with 30 mg/kg of sodium nitrite; with 10 g/kg of grape seed extract; with 10 g/kg of chestnut extract. Drying process: 4 days, 28 °C, 85% RH. Ripening process: 21 days, 13 °C, 70% RH.	Formulas without nitrite revealed lower pH values. *Lactobacillaceae* were significantly more present in chestnut extract formula. Although all three formulas showed significant differences, natural extracts did not present drastic changes in the prokaryotic community or other physicochemical parameters.	Grape seed extract presented less antioxidant capacity than sodium nitrite. Products without nitrite were less red and dark when compared with control.	[99]
Fermented smoked sausage	Lithuania	Lyophilized celery, with the addition of *S. xylosus* or *S. xylosus* and *P. pentosaceus* mixture	Evaluate lyophilized celery as possible substitute for both nitrite and nitrate regarding quality and safety.	6 formulas: with 150 mg/kg of sodium nitrate and *S. xylosus*; with 150 mg/kg of sodium nitrate and *S. xylosus* and *P. pentosaceus* mixture; with 150 mg/kg d of sodium nitrite and *S. xylosus*; with 150 mg/kg of sodium nitrite and *S. xylosus* and *P. pentosaceus* mixture; with lyophilized celery and *S. xylosus*; with lyophilized celery and *S. xylosus* and *P. pentosaceus* mixture. Fermentation and drying processes: 14 days. Initial temperature 24 °C, 92% RH and gradual decrease until 15 °C, 76% RH (9th–10th day) and then remained constant. Smoking process: cold smoking at 96, 120 e 168 h.	Sausages with *S. xylosus* revealed less residual nitrate content than those with the addition of *S. xylosus* and *P. pentosaceus* mixture. Starter culture with *S. xylosus* and *P. pentosaceus* mixture presented positive effect in reddish color, higher than the effect of the *S. xylosus* culture; Lyophilized celery might have potential as an alternative for nitrites and nitrates if conjugated with starter cultures that help control fermentation and ripening processes.	The reddish color was less intense in sausages with lyophilized celery’s addition.	[100]
Fermented cured sausage	Brazil	Radish and beetroot powder, with the addition of *Staphylococcus carnosus*	Evaluate the effect on the development of cured characteristics during ripening and storage in physicochemical and microbiological parameters.	6 formulas: control with 150 mg/kg of sodium nitrate and 150 mg/kg of sodium nitrite; control without nitrate nor nitrite; with 0.5% of beetroot powder; with 1% of beetroot powder; with 0.5% of radish powder; with 1% of radish powder. Formulas with beetroot and radish powder were complemented with a starter culture *S. carnosus*. Fermentation and drying process: 1st day, 25 °C/95%; 2nd day, 24 °C/93%; 3rd day, 23 °C/91%; 4th day, 22 °C/89%; 5th day, 21 °C/87%; 6th day, 20 °C/87%; 7th day, 18 °C/85%; 8th to 35th day, 15 °C/75%. The ripening process ended when a_w_ < 0.91. Storage: packed in vacuum, 60 days, 5 °C.	Vegetable powder addition lowered humidity and a_w_, increasing the weight loss of the sausages. Of all formulas studied, the addition of 1% radish powder was the best alternative to nitrite, considering the following parameters: pH, colour, residual nitrate and nitrite content, and LAB development.	The main negative impact of beetroot’s powder addition was its effect in sausage’s color.	[91]

RH: relative humidity; LAB: lactic acid bacteria.

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
