# Peer review of "Nitrate Is Nitrate: The Status Quo of Using Nitrate through Vegetable Extracts in Meat Products"

_foods, 2021, doi:10.3390/foods10123019_

Round 1
Reviewer 1 Report
This review provides interesting data. However, there are some parts that are difficult to understand
and can be improved.
L28 What is “fragile equilibrium”?
L141,L143 Recheck citation style
L158-159 This sentence should be carefully considered. There are some cases that the tradition
production add more nitrate/nitrite than large scale commercialization as there is no strict control of
the regulation as compared with large commercial product.
Fig1 The font is too small.
Table 1 caption should add …”in several countries.”
Please recheck the style of using “,” and “.” For number with decimal.
Table 2 Please revise the caption
What does the sigma symbol mean?
L298 Please add recent research about attempt to reduce residual nitrite in food by active packaging
as “Other methods to reduce residual nitrite in meat was the use of active packaging with
incorporated nitrite which gave less amount of nitrite than direct addition into foods, while
preserving redness
L317 There were recent utilization of plant extract incorporated active packaging to replace nitrite in
order to extend meat shelf-life and delayed microbial growth.
Page 14 [99] Right column, does this mean the natural extract gave the best quality reservation?
Page 15 [91] Increase the weight loss… so it is not good to being used?
Conclusions
This part should be revised as this is the review. Conclusion should not repeat the discussion but is
should provide authors critics/recommendation or highlight of the use of nitrate/nitrite or future
trend in terms of industry or academic.
Author Response
Responses to REVIEWER 1
The authors acknowledge all the suggestions and comments from reviewers that permit the improvement of this paper.
This review provides interesting data. However, there are some parts that are difficult to understand and can be improved.
Reply: The text was revised and improved.
L28 What is “fragile equilibrium”?
Reply: We acknowledge the reviewers comment. The word “fragile” introduced some confusion in the sense of the phrase. The term was replaced by “equilibrium”.
L141,L143 Recheck citation style –
Reply: The citation style was corrected.
L158-159 This sentence should be carefully considered. There are some cases that the tradition production add more nitrate/nitrite than large scale commercialization as there is no strict control of the regulation as compared with large commercial product.
Reply: We acknowledge the reviewers comment. The idea of this sentence was related to the “inevitability” of using nitrite in big industries, when compared to small producers. It might not be the case of the residual amount itself, but the use or not. The sentence and the idea was clarified.
Fig1 The font is too small. ~
Reply: the font was increased.
Table 1 caption should add …”in several countries.
Reply: the change was done
Please recheck the style of using “,” and “.” For number with decimal.
Reply: the check was done.
Table 2 Please revise the caption.
Reply: the change was done
What does the sigma symbol mean?
Reply: Sigma symbol mean micro (µg = micrograms).
L298 Please add recent research about attempt to reduce residual nitrite in food by active packaging as “Other methods to reduce residual nitrite in meat was the use of active packaging with incorporated nitrite which gave less amount of nitrite than direct addition into foods, while preserving redness
Reply: the sentence was rephrased with addition of recen work regarding active packaging
L317 There was recent utilization of plant extract incorporated active packaging to replace nitrite in order to extend meat shelf-life and delayed microbial growth.
Reply: the suggestion was taken in consideration and added to the body text.
Page 14 [99] Right column, does this mean the natural extract gave the best quality reservation?
Reply: In fact, nitrite-free products of this study revealed lower pH values, probably due to the higher levels of Lactobacillaceae, and it is known that a low pH is associated with higher safety. However, the authors do not state that any of the studied extracts revealed the best quality preservations. Instead, they suggest that these extracts might substitute nitrite, precisely because “natural extracts did not drastically alter the prokaryotic community or the other chemical/physical parameters”. In other words, authors considered the results quite similar.
Page 15 [91] Increase the weight loss… so it is not good to being used?
Reply: According to the authors, control samples showed weight losses around 35%, meanwhile formulas with radish and beetroot presented values between 41.46% and 42.74%. This difference was not considered an issue by authors. Instead, they highlight that lower aw might be important for product’s safety, especially in formulas without nitrite. We believe, considering available data, that in this case industry would have to choose between a product without synthetic nitrite that might fulfil consumers desires or a product with less weight loss and consequently higher profit.
Conclusions
This part should be revised as this is the review. Conclusion should not repeat the discussion but is should provide authors critics/recommendation or highlight of the use of nitrate/nitrite or future trend in terms of industry or academic.
Reply: We acknowledge the reviewers comment. We reorganized the main ideas in the conclusions, to fulfill the suggests of both reviewers. Our point of view is now clearly stated in the conclusion, and a synthesis of the main research findings was presented.

Reviewer 2 Report
The present paper has in attention the use of nitrate and nitrite in meat products.
For the readers' convenience, the authors are kindly requested to consider the following recommendations:
Improve the quality of the figure and move it closer to the paragraph referring to it.
Figure 2. Check the title of this figure and extend the discussion of the presented results in the manuscript
Section 4: Consider moving the table with the most used vegetables next to the paragraph citing this table. It is commendable to insert a column in the table, referring to the main disadvantages of the plant usages.
Conclusions: This section should be improved and presented following the main findings presented in the review.
Author Response
The authors acknowledge all the suggestions and comments from reviewers that permit the improvement of this paper.
Responses to REVIEWER 2
The present paper has in attention the use of nitrate and nitrite in meat products.
For the readers' convenience, the authors are kindly requested to consider the following recommendations:
Improve the quality of the figure and move it closer to the paragraph referring to it. Reply: the change was done.
Figure 2. Check the title of this figure and extend the discussion of the presented results in the manuscript
Reply: We believe reviewer meant “Table 2”. Title was checked. Table results are now briefly discussed in the manuscript.
Section 4: Consider moving the table with the most used vegetables next to the paragraph citing this table. It is commendable to insert a column in the table, referring to the main disadvantages of the plant usages.
Reply: Table 3 is now cited in section 5. We believe it is as near as possible. We appreciate the recommendation regarding disadvantages, therefore we reformulated Table 3 in order to include some disadvantages referred by authors.
Conclusions: This section should be improved and presented following the main findings presented in the review.
Reply: We acknowledge the reviewers comment. We reorganized the main ideas in the conclusions, to fulfil the suggestions of both reviewers. Our point of view is now clearly stated in the conclusion, and a synthesis of the main research findings was presented.

Round 2
Reviewer 1 Report
The manuscript has been improved. It will be a good reference in the field.
Reviewer 2 Report
The authors have addressed all of the reviewer's comments.